# Genetic Alterations and Risk Factors for Recurrence in Patients with Non-Small Cell Lung Cancer Who Underwent Complete Surgical Resection

**DOI:** 10.3390/cancers15235679

**Published:** 2023-11-30

**Authors:** Hwa Kyung Park, Yoo Duk Choi, Ju-Sik Yun, Sang-Yun Song, Kook-Joo Na, Joon Young Yoon, Chang-Seok Yoon, Hyung-Joo Oh, Young-Chul Kim, In-Jae Oh

**Affiliations:** 1Lung Cancer Center, Chonnam National University Hwasun Hospital, Gwangju 58128, Republic of Korea; melong526@naver.com (H.K.P.); drydchoi@chonnam.ac.kr (Y.D.C.); jusikyun@chonnam.ac.kr (J.-S.Y.); sysong@jnu.ac.kr (S.-Y.S.); kjna1125@hanmail.net (K.-J.N.); yjyowl@gmail.com (J.Y.Y.); ycs5151@gmail.com (C.-S.Y.); ohj4250@naver.com (H.-J.O.); kyc0923@chonnam.ac.kr (Y.-C.K.); 2Department of Internal Medicine, Chonnam National University Medical School, Gwangju 61469, Republic of Korea; 3Department of Pathology, Chonnam National University Hospital, Gwangju 61469, Republic of Korea; 4Department of Thoracic and Cardiovascular Surgery, Chonnam National University Hwasun Hospital, Gwangju 58128, Republic of Korea; 5Department of Internal Medicine, Chonnam National University Hospital, Gwangju 61469, Republic of Korea

**Keywords:** epidermal growth factor receptor, gene, non-small cell lung cancer, recurrence, surgery

## Abstract

**Simple Summary:**

We investigated the prevalence of genetic alterations and their association with epidermal growth factor receptor (EGFR) mutations and prognosis in early-stage non-small cell lung cancer (NSCLC) after curative resection. The results showed that the prevalence of EGFR mutations, anaplastic lymphoma kinase (ALK) rearrangements, and ROS proto-oncogene 1 (ROS1) fusion were 43.0%, 5.7%, and 1.6%, respectively. Patients with EGFR-mutant NSCLC had a higher risk of recurrence than those without EGFR mutations. Additionally, EGFR mutations were related to a high proportion of distant metastases and a higher risk of central nervous system recurrence.

**Abstract:**

A definitive surgical resection is the preferred treatment for early-stage non-small cell lung cancer (NSCLC). Research on genetic alterations, including epidermal growth factor receptor (EGFR) mutations, in early-stage NSCLC remains insufficient. We investigated the prevalence of genetic alterations in early-stage NSCLC and the association between EGFR mutations and recurrence after a complete resection. Between January 2019 and December 2021, 659 patients with NSCLC who underwent curative surgical resections at a single regional cancer center in Korea were recruited. We retrospectively compared the clinical and pathological data between the recurrence and non-recurrence groups. Among the 659 enrolled cases, the median age was 65.86 years old and the most common histology was adenocarcinoma (74.5%), followed by squamous cell carcinoma (21.7%). The prevalence of EGFR mutations was 43% (194/451). Among them, L858R point mutations and exon 19 deletions were 52.3% and 42%, respectively. Anaplastic lymphoma kinase (ALK) rearrangement was found in 5.7% of patients (26/453) and ROS proto-oncogene 1 (ROS1) fusion was found in 1.6% (7/441). The recurrence rate for the entire population was 19.7%. In the multivariate analysis, the presence of EGFR mutations (hazard ratio (HR): 2.698; 95% CI: 1.458–4.993; *p* = 0.002), stage II (HR: 2.614; 95% CI: 1.29–5.295; *p* = 0.008) or III disease (HR: 9.537; 95% CI: 4.825–18.852; *p* < 0.001) (vs. stage I disease), and the presence of a pathologic solid type (HR: 2.598; 95% CI: 1.405–4.803; *p* = 0.002) were associated with recurrence. Among the recurrence group, 86.5% of the patients with EGFR mutations experienced distant metastases compared with only 66.7% of the wild type (*p* = 0.016), with no significant difference in median disease-free survival (52.21 months vs. not reached; *p* = 0.983). In conclusion, adjuvant or neoadjuvant targeted therapy could be considered more actively because EGFR mutations were identified as an independent risk factor for recurrence and were associated with systemic recurrence. Further studies on perioperative therapy for other genetic alterations are necessary.

## 1. Introduction

In 2020, lung cancer was reported to be the leading cause of cancer-related death worldwide and the second most commonly diagnosed cancer [1]. Approximately 48% of lung cancer cases in the U.S. are diagnosed in the early stage [2,3]. Patients with early-stage lung cancer have a better 5-year survival rate compared with those with advanced-stage (62.8% vs. 8.2%); therefore, it is crucial to diagnose lung cancer early to ensure the best chance of preventing recurrence [4].

A definitive surgical resection is the preferred treatment for early-stage non-small cell lung cancer (NSCLC), followed by adjuvant chemotherapy if necessary, depending on the pathological stage [5]. A retrospective analysis revealed that patients with stage I NSCLC who underwent surgical resection had a 5-year survival rate ranging from 40% to 97% [6]. According to a real-world study conducted in the U.S., only approximately 40% of patients with stage IB–IIIA NSCLC received adjuvant chemotherapy after surgical resection and 23% of them experienced disease recurrence after 1 year [7]. Therefore, it is crucial to predict recurrence after surgical resection and select patients who are likely to see the greatest benefit from an adjuvant therapy. The decision to use an adjuvant therapy is mainly based on the TNM stage, which has been expanded to include pathologic findings in the updated guidelines. Molecular tests, epidermal growth factor receptor (EGFR) mutations, anaplastic lymphoma kinase (ALK) rearrangements, and programmed death-ligand 1 (PD-L1) expression have also been investigated in patients with early-stage NSCLC [8].

Smoking increases the risk of some, but not all, types of lung cancer. Some patients with lung cancer harboring EGFR mutations do not smoke. EGFR mutations are the most common driver mutation in advanced NSCLC in East Asia, with a prevalence of approximately 20–64% [9]. In a large-scale data analysis conducted in China, the EGFR mutation rate in the early stage was 53.6%, which was comparable with the rate observed in the advanced stage (51.4%; *p* = 0.379). The proportion of EGFR mutations was higher in stage IA disease than in stages IIB and IIIA. In a retrospective multi-center analysis, the presence of EGFR mutations was significantly related to disease recurrence, along with lymphovascular invasion, intrapulmonary metastasis, and lymph-node metastasis. The disease-free survival (DFS) of patients with early-stage NSCLC who had EGFR mutations was shorter than that of patients without EGFR mutations, even if they shared the same disease stage [10]. Another study showed that the presence of EGFR mutations was a risk factor for distant metastasis in early-stage lung cancer [11]. EGFR mutations, an unfavorable factor for recurrence, warrant careful consideration. However, studies on the prevalence of EGFR mutations in patients with early-stage NSCLC are limited. Despite this, many studies on EGFR tyrosine kinase inhibitors (EGFR-TKIs) have been expanded to perioperative therapy from palliative therapy in the advanced stage [12,13,14,15].

Osimertinib, a third-generation EGFR-TKI, demonstrated a significant improvement in DFS compared with a placebo in patients with stage IB–IIIA EGFR-mutated NSCLC who underwent complete resection [15]. According to the updated follow-up data of the ADAURA study, the 4-year DFS rate was 73% in the osimertinib group and 38% in the placebo group (overall hazard ratio (HR) for DFS: 0.23; 95% confidence interval (CI): 0.18 to 0.30) [15]. Based on these successful results, osimertinib is recommended as an adjuvant treatment for resected EGFR-mutant stage IB–IIIA NSCLC [16,17]. A previous systemic review revealed that adjuvant EGFR-TKIs prolonged DFS in patients with EGFR mutations regardless of the mutation subtype, but could not prolong overall survival (OS) [18]. Given these suboptimal results, platinum-doublet chemotherapy is still preferred as an adjuvant therapy in NSCLC [8]. Additionally, it has been suggested that natural polyphenols have a potential contribution to lung cancer treatment [19].

Despite much progress during the last decade, including targeted therapy and immunotherapy in advanced stages, the prognosis of NSCLC remains dismal. Therefore, more information is needed on patients with early-stage disease, especially in Asian populations, which tend to have a large number of non-smokers. In this study, we investigated the prevalence and importance of genetic alterations in patients with early-stage NSCLC who underwent definitive surgery in a single regional cancer center of Korea. 

## 2. Materials and Methods

### 2.1. Data Collection

We collected data on 1095 patients who underwent surgery at the Lung Cancer Center, Chonnam National University Hwasun Hospital (CNUHH), from 1 January 2019 to 31 December 2021 and retrospectively reviewed their electronic medical records (Figure 1). First, 205 patients were excluded for the following reasons: 34 patients received the second operation after their first operation before 2019; 2 patients were diagnosed with benign disease; 101 patients underwent diagnostic surgical resections at stage IV or stage III, which were indicated for chemotherapy or concurrent chemoradiotherapy; 1 patient underwent an open and closure operation; 1 patient underwent their first operation at another hospital; 11 patients were diagnosed with small cell lung cancer; 24 patients did not undergo positron emission computed tomography (PET-CT), so distant metastases were not evaluated; 13 patients were diagnosed with a disease other than lung cancer; 3 patients were diagnosed with double-primary malignancies; and 15 patients received an incomplete resection. We also excluded 227 patients who did not undergo mediastinal lymph-node dissection. Among the patients who underwent a definitive surgical resection with mediastinal lymph-node dissection, we excluded four patients who were diagnosed at stage 0. 

We collected the clinical and pathological information of 659 patients. The collected data included sex, age, history of smoking or second-hand smoking, comorbidities (e.g., hypertension, diabetes mellitus, coronary disease, peripheral vascular disease, liver disease, history of pulmonary tuberculosis, interstitial lung disease, chronic obstructive pulmonary disease, or other malignancies), family history of lung cancer, Eastern Cooperative Oncology Group Performance Score (ECOG PS), parameters of a pulmonary function test (e.g., FEV_1_, FVC, and DL_CO_), serum carcinoembryonic antigen (CEA), serum pro-gastrin releasing peptide (proGRP), serum Cyfra21-1, histologic type, histologic subtype, pathologic stage, presence of EGFR mutations, anaplastic lymphoma kinase (ALK) rearrangements, ROS proto-oncogene 1 (ROS1) fusion, programmed death-ligand 1 (PD-L1) expression, initial therapy, operation type, surgical approach type, date of surgery, tumor location, presence of visceral pleural invasion, lymphatic invasion, vascular and neural invasion, implementation of an adjuvant therapy, recurrence state, date of recurrence, survival state, and date of last follow-up. The recurrence date was set as the date on which the treatment for recurrence began.

This study was conducted in accordance with the Declaration of Helsinki (as revised in 2013). This study was approved by the Institutional Review Board of Chonnam National University Hwasun Hospital (CNUHH-2023-164). The requirement for patient consent was waived due to the retrospective nature of the study and the use of anonymized clinical data for the analysis.

### 2.2. Detection of EGFR Mutations

DNA was isolated from formalin-fixed paraffin-embedded (FFPE) tumor tissue using a Gene All Tissue DNA Purification Kit (General Biosystems, Seoul, Republic of Korea) according to the manufacturer’s protocol. The obtained DNA was eluted in 50 μL of an elution buffer and the concentration and purity of the extracted DNA were assessed by spectroscopy using a NanoDrop spectrophotometer (NanoDrop Technologies Inc., Wilmington, DE, USA). We then detected EGFR gene mutations with a real-time polymerase chain reaction (PCR) using a PNA Clamp Mutation Detection Kit (Panagene Inc., Daejeon, Republic of Korea). All reactions were conducted in 20 μL volumes using template DNA, a primer, a PNA probe set, and a fluorescence PCR master mix. Real-time PCR was performed using a CFX 96 (BioRad Laboratories Inc., Hercules, CA, USA). PCR cycling conditions were set as follows: UDG incubation and pre-denaturation time for 1 cycle, at 50 °C for 20 min, 95 °C for 15 min, and first 3-step cycles for 15 cycles, at 95 °C for 30 s, 70 °C for 20 s, 63 °C for 1min, second 3-step cycles for 35 cycles, at 95 °C for 10 s, 53 °C for 20 s, 73 °C for 20 s, melting curve analysis cycle for 1 cycle at 95 °C for 15 min, 35 °C for 5 min, 35 °C to 75 °C increment 0.5 °C for 30 s. The pooled sensitivity and specificity of the PNA clamp methods were 93% and 100%, respectively [20,21,22].

### 2.3. Detection of ALK Rearrangements

We used an ALK break-apart probe (Abbott Vysis ALK Break Apart FISH Probe Kit; Abbott Molecular, Abbott Park, IL, USA) to investigate ALK rearrangements in FFPE operative tissue. These probes have a dual-color system, which realizes the ALK break-point cluster with red and green signals. The detection of ALK rearrangements was performed according to the method reported in a previous study. In the FFPE tissue with a maintained ALK gene, the probes were visualized by the point of fusion of red and green signals. However, in the FFPE tissue with a disrupted ALK locus, the red and green signals split and consequently showed a split pattern. We defined ALK-positive tissues as those in which more than 15% of cancer cells showed split or deleted-split patterns [23].

### 2.4. Detection of ROS1 Fusion

ROS1 fusion was detected with a PCR using a ROS1 fusion gene detection kit (Amoy Diagnostics Co., Ltd., Xiamen, China). The total RNA isolated from the FFPE tissue from each specimen was used to detect ROS1 fusion. Reverse transcription was performed using total RNAs and the conditions were as follows: 42 °C for 1 h and 95 °C for 5 min. The resulting complementary DNA (cDNA) solutions were used for multiplex qRT-PCR and the ROS1 fusion gene mRNA was detected using qRT-PCR. The PCR procedure was as follows: initial denaturation at 95 °C for 5 min, ensuring specificity at 95 °C for 25 s, 64 °C for 20 s, and 72 °C for 20 s; and data collection for 31 cycles of 93 °C for 25 s, 60 °C for 35 s, and 72 °C for 20 s. The fusion patterns were investigated according to a previous study [24]. The quantitative analysis was based on fusion fluorescence signals. Test responses that achieved a Ct value of fewer than 30 cycles were defined as ROS1-fusion-positive.

### 2.5. Statistical Analysis

Overall survival (OS) was defined as the period from the date of operation to death and DFS was defined as the period from the date of operation to recurrence. Genetic alterations were defined as cases in which preoperative or surgical tissues showed EGFR mutations, ALK rearrangements, or ROS1 fusion. If both preoperative and surgical tissues were available, PD-L1 expression was determined based on the surgical tissues using an SP263 antibody. We defined the pathologic stage according to the 8th edition of the AJCC Cancer Staging Manual. The data cut-off date was 12 July 2023.

We compared the clinical and pathological backgrounds of patients with and without recurrence and compared the genetic alterations between locoregional recurrence and distant recurrence in patients who experienced recurrence. We used the Mann–Whitney U test to analyze the continuous variables and the chi-squared test or Fisher’s exact test were used for the categorical variables. To predict the risk factors for recurrence after surgical resection, variables with a *p*-value < 0.25 in the univariate logistic regression were analyzed using a multivariate logistic regression. Kaplan–Meier analyses and Cox proportional hazard modeling were used to investigate the correlation between EGFR mutations and DFS or OS. All statistical analyses were performed using SPSS Statistics for Windows, version 27 (IBM Corp., Armonk, NY, USA). A *p*-value < 0.05 was considered to be statistically significant.

## 3. Results

### 3.1. Baseline Characteristics of the Overall Patients

We included 659 patients with NSCLC who underwent complete resections with lymph-node dissection from January 2019 to December 2021 (Table 1). The majority of patients (87.6%) underwent a lobectomy. Overall, 19 patients (2.9%) had stage IA1, 99 patients (15.0%) had stage IA2, 134 patients (20.3%) had stage IA3, 169 patients (25.6%) had stage IB, 38 patients (5.8%) had stage IIA, 89 patients (13.5%) had stage IIB, 98 patients (14.9%) had stage IIIA, and 13 patients (2.0%) had stage IIIB. Out of 451 patients, 194 (43%) had EGFR mutations, 26 out of 453 patients (5.7%) had ALK rearrangements, and 7 out of 441 patients (1.6%) had a ROS1 fusion. Of the 479 patients, 295 (61.6%) had a PD-L1 expression < 1%, 113 patients (23.6%) had a PD-L1 expression 1–49%, and 71 (14.8%) had a PD-L1 expression ≥ 50%. The most commonly diagnosed histological type was adenocarcinoma (73.7%), followed by squamous cell carcinoma (22.2%). Overall, 243 patients (36.9%) received an adjuvant therapy after surgical resection. Among them, 236 patients (97.1%) received platinum-doublet chemotherapy, 1 patient (4.1%) received a target therapy, and 6 patients (2.5%) received concurrent chemoradiotherapy. 

In total, 130 patients (19.7%) experienced disease recurrence. We divided the patients into two groups: the non-recurring group and the recurring group. In the non-recurring group, 377 patients (71.3%) had stage I disease, 100 patients (18.9%) had stage II disease, and 52 patients (9.8%) had stage III disease. In the recurring group, 44 patients (33.8%) had stage I disease, 27 patients (20.8%) had stage II disease, and 59 patients (45.4%) had stage III disease (*p* < 0.001). The proportion of EGFR mutations was similar in the non-recurring and recurring groups (41.2% vs. 49.1%; *p* = 0.151). In both groups, most patients underwent a lobectomy (87.1% in the non-recurrence group and 89.2% in the recurrence group).

### 3.2. Prevalence of Genetic Alterations

In the patients with EGFR-mutated NSCLC, the majority had stage I disease (64.4% with stage I disease compared with 17% with stage II disease and 18.6% with stage III disease; *p* = 0.047). In patients with ALK-positive NSCLC, most patients had stage I or III disease (46.2% with stage I or III disease compared with 7.7% with stage II disease; *p* = 0.002). The same trend was observed for ROS1 fusion, although it did not reach a statistical significance (42.9% with stage I or III disease compared with 14.3% with stage II disease; *p* = 0.283) (Figure 2a). Among the EGFR-mutant patients, the most frequently observed mutation was L858R, which was noted in 101 patients (52.3%), followed by Ex19del, which was observed in 79 patients (40.9%). Moreover, 2 patients (1.0%) had both L858R and Ex19del. Other rare mutations were noted in 11 patients (5.7%) (Table 2). The higher the expression of PD-L1, the higher the proportion of a higher stage, but there was no statistical significance (*p* = 0.063) (Figure 2b).

According to the prevalence based on the day of surgery, only 42 patients (6.4%) were found to have genetic alterations before surgery. Preoperative biopsy sites were the primary lung mass (41 cases) or metastatic lymph node (1 case). EGFR mutations were found in 14 patients (33.3%), including 6 cases of L858R, 6 cases of Ex19del, 1 case of G719S, and 1 case of an exon 20 insertion. There was one case each of an ALK rearrangement and a ROS1 fusion.

### 3.3. Prognostic Factors Associated with Disease Recurrence 

The median follow-up duration for all patients was 31.54 months (95% CI: 30.10–32.98). Death, hopeless discharge, or loss to follow-up occurred in 87 patients (13.2%) and the median DFS was 30.52 months (95% CI: 29.21–31.83). Recurrence after definitive surgery occurred in 130 patients (19.7%). Out of the 106 patients investigated for the presence of EGFR mutations and among those with recurrence, 52 patients (49.1%) had EGFR mutations. 

First, we analyzed the risk factors for recurrence after definitive surgery in all patients. In the univariate analysis, being male (vs. female), an ECOG PS of 1 (vs. PS 0), stage II or III (vs. stage I) disease, the presence of ALK rearrangements, VPI, lymphatic or vascular invasion, and the pathologic subtype (presence of a micropapillary or solid type, and absence of lepidic type) were associated with disease recurrence. In the multivariate analysis, stage II or III (vs. stage I) disease, the presence of EGFR mutations, and the pathologic subtype (presence of a solid type) were associated with disease recurrence (Figure 3a). However, there was no significant difference in DFS between the EGFR wild-type group and the mutant group (Figure 3b).

In stage IB–IIIA disease, recurrence occurred in 109 patients (27.7%). In the univariate analysis, stage III (vs. stage I) disease, the presence of EGFR mutations and ALK rearrangements, lymphatic or vascular invasion, and the pathologic subtype were associated with recurrence. In the multivariate analysis, stage III (vs. stage I) disease and the pathologic subtype (absence of acinar or mucinous type) were associated with recurrence (Figure 3c). However, there was no significant difference in DFS between the EGFR wild-type group and the mutant group (Figure 3d).

### 3.4. Association between EGFR Mutations and Type of Recurrence

In patients who experienced recurrence, locoregional recurrence occurred in 37 patients (28.5%), whereas distant metastases occurred in 93 patients (71.5%). The most commonly recurring extrathoracic location was the bones (30 patients; 23.1%), followed by the central nervous system (CNS) (23 patients; 17.7%). The CNS was the most common metastatic location for patients with EGFR-mutated NSCLC and the presence of EGFR mutations was a risk factor for CNS metastasis (Table 3). Among the overall patients, 86.5% of EGFR-mutant patients experienced distant metastases, which was higher than the rate observed in EGFR-wild patients (66.7%; *p* = 0.016). Additionally, in patients with stage III disease, a higher proportion of those with EGFR mutations experienced distant metastases compared with those without EGFR mutations (92% vs. 68%; *p* = 0.034). However, there was no significant difference in distant metastases between EGFR-mutant patients and EGFR-wild patients with stage I (86.7% vs. 68.4%, *p* = 0.257) and stage II (75.0% vs. 60.0%, *p* = 0.652) disease (Figure 4a). Although the proportion of distant metastases was higher in the EGFR-mutant group than in the wild-type group over all patients, there was no significant difference in OS (Figure 4b).

## 4. Discussion

In this study, we evaluated the prevalence of genetic alterations in patients with early-stage NSCLC and identified a risk factor for disease recurrence. EGFR mutations are more common in Asians than in Caucasians [25,26]. It has been noted that EGFR mutations occur in approximately 30–50% of Asian patients with advanced NSCLC. The prevalence of EGFR mutations in early-stage NSCLC was found to be similar to that in advanced-stage NSCLC according to a retrospective analysis published in China [9,26]. We found that approximately two-thirds of all patients were tested for genetic alterations, among whom, 43% had EGFR mutations. The prevalence of EGFR mutations in this study was similar to that of Asians in previous studies [27,28].

The significance of EGFR mutations associated with prognosis of early-stage NSCLC has been reported, along with various factors such as age, histologic pattern, pathologic stage, and presence of lymphovascular invasion [29,30,31,32,33]. Saw et al. reported an association between EGFR mutations and recurrence in resected NSCLC for stage IA to IIIA disease. The 5-year OS was better, but the recurrence rate was higher in patients with EGFR-mutated NSCLC than in those with the wild type [32]. 

EGFR mutations have been reported to be harbored more frequently in patients with a lepidic component than those without [34,35]. A lepidic component is a favorable prognostic factor and usually has a part-solid appearance in computed tomography [36,37]. The volume and diameter of ground glass opacity have been shown to be correlated with EGFR mutations, while the presence of EGFR mutations has been shown to be correlated with the growth of ground glass nodules [38,39]. Moreover, a previous study revealed that EGFR mutations, especially L858R mutations, increased the cell invasion ability in lung adenocarcinoma [40]. Additionally, it has been theorized that circulating tumor cells increase in the draining pulmonary vein during surgical resection, which is particularly significant in patients with lymphatic invasions [41]. Taken together, these findings infer that EGFR mutations are associated with recurrence after surgical resection due to the high invasiveness. 

Although EGFR mutations did not impact the DFS and OS in this study, they were associated with a higher risk of recurrence and the type of metastasis. Among all patients, the proportion of distant metastases was higher in the EGFR-mutant group than in the EGFR-wild group and the same trend was observed in patients with stage III NSCLC. In addition, the presence of EGFR mutations was a risk factor for CNS metastasis, which represents a major concern after definitive surgery in early-stage NSCLC [42,43]. It has long been known that patients with brain metastases have particularly unfavorable prognoses [44]. Despite the poor prognoses of patients who experience recurrence after surgical resection, fewer than 50% of patients who underwent definitive surgery for early-stage NSCLC received an adjuvant therapy [7,45]. Furthermore, the proportion of patients with EGFR-positive NSCLC who receive adjuvant chemotherapy, which is not a targeted therapy, after definitive resection is still high [46,47]. The situation was similar at our center. We discovered that 23% of patients received an adjuvant therapy after curative resection, among whom, only one patient received a target therapy. However, platinum-based adjuvant chemotherapy has been shown to be ineffective in preventing CNS metastasis [48].

The recently updated guidelines recommend an investigation of the EGFR mutation status and ALK rearrangements for adjuvant therapies in resectable NSCLC [8]. With many efforts to reduce recurrence and improve survival in early-stage NSCLC, a new generation of EGFR-TKIs has opened up a new era of adjuvant therapies after definitive surgery in recent years. Icotinib, a first-generation EGFR-TKI, improved DFS compared with chemotherapy in patients with stage II–IIIA NSCLC after resection [13]. Gefitinib, another first-generation EGFR-TKI, improved DFS in patients with completely resected stage II–IIIA EGFR-mutated NSCLC [12]. According to the recently updated results of the ADAURA study, adjuvant osimertinib for EGFR-mutated NSCLC after complete resection improved not only DFS but also 5-year OS in patients with stage IB–IIIA disease. As a result of an analysis by the stage, the 5-year OS was significantly different only in stage IIIA disease [15]. Consequently, it is important to gather more individualized information about patients. The prognosis varies depending on the genetic alterations and pathologic subtypes, even within the same stage. It is crucial to select appropriate high-risk patients and determine the adjuvant therapy required to prevent recurrence in early-stage lung cancer and improve survival. Furthermore, some promising results of neoadjuvant targeted therapy are being reported [49]. Given the high rate of distant metastases after resection in EGFR-mutated NSCLC and the benefit of EGFR-TKIs on survival, it seems important to detect genetic alterations before surgery. In this study, they were only found in 6.4% of cases before surgery. A CT-guided core needle biopsy, which is a commonly used biopsy modality, has a sensitivity of 90.07% and accuracy of 98.87%, with an AUC of 0.952. Additionally, it is rare for serious complications to occur after a CT-guided core needle biopsy [50]. Along with this safe and accurate diagnostic tool, if genetic alterations can be found with a more sensitive method such as a liquid biopsy, the prognosis of lung cancer might be improved with neoadjuvant targeted therapies.

This study had several limitations that warrant discussion. First, this was a single-center retrospective study. Although the proportion of genetic alterations and prognosis of patients followed the same trend as those in previous studies, the characteristics of the patients in this study may not fully represent all Koreans or Asians. Second, an analysis of genetic alterations was not performed in all patients. The presence of genetic alterations was mainly investigated in patients with adenocarcinoma; in other types of NSCLC, including squamous cell carcinoma, the expression of PD-L1 was investigated. Third, other factors such as sex, age, and pathologic subtype were not adjusted for when analyzing the association between metastatic organs and the presence of EGFR mutations. 

## 5. Conclusions

In this study, we showed that the prevalence of EGFR mutations, ALK rearrangements, and ROS1 fusion were 43.0%, 5.7%, and 1.6%, respectively, in patients with NSCLC who underwent complete resections. EGFR mutations were an independent risk factor for recurrence, along with stage II/III disease and a solid pathologic subtype, and were associated with a high rate of distant metastases and a higher risk of CNS recurrence compared with wild-type EGFR. Therefore, adjuvant or neoadjuvant targeted therapies should be more actively considered in patients with EGFR mutations who have undergone curative surgery. Further studies on perioperative therapies for other genetic alterations are also needed and it is important to establish more sensitive methods such as liquid biopsies to predict recurrence and provide optimal therapies.

## Figures and Tables

**Figure 1 cancers-15-05679-f001:**
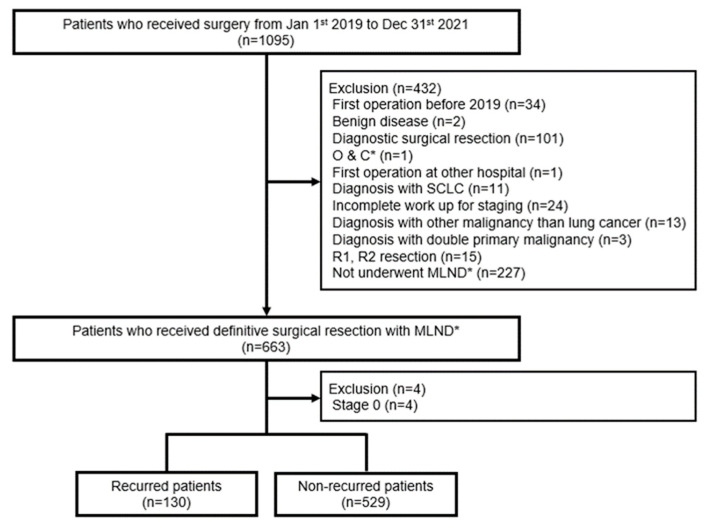
Flowchart of patient enrollment. * O & C: open and closure; MLND: mediastinal lymph-node dissection.

**Figure 2 cancers-15-05679-f002:**
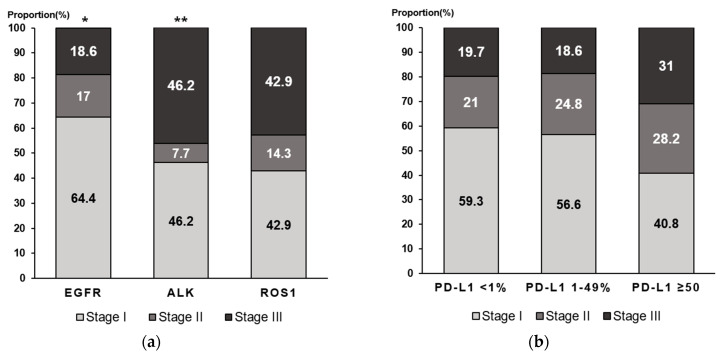
Proportion of genetic alterations. (**a**) Bar graph showing the percentage of genetic alterations by stage over all patients. Among the patients with EGFR mutations, the majority had stage I disease. In patients who were ALK-positive, the majority had stage I and III disease. (**b**) Bar graph showing that the higher the PD-L1 expression, the higher the proportion of a high stage, but this was not statistically significant (*p* = 0.063). * *p* < 0.05; ** *p* < 0.01. EGFR: epidermal growth factor receptor; ALK: anaplastic lymphoma kinase; PD-L1: programmed death-ligand 1.

**Figure 3 cancers-15-05679-f003:**
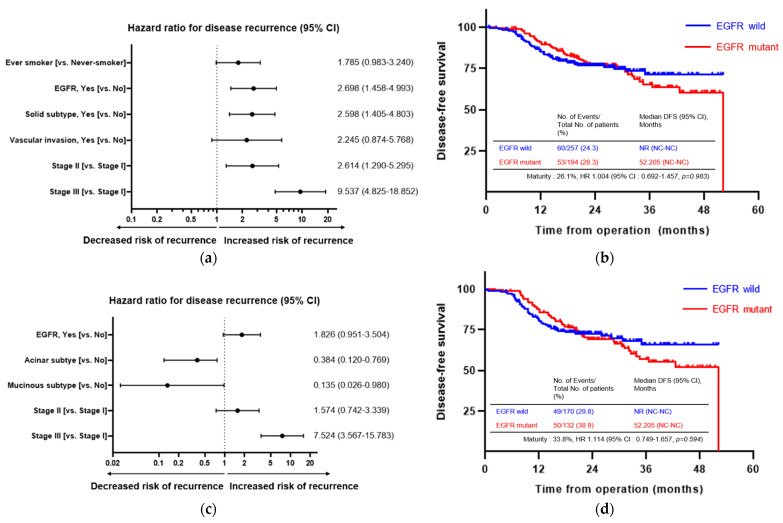
Risk factors for disease recurrence. (**a**) Forest plots for risk factors of disease recurrence over all patients. (**b**) Kaplan–Meier survival curve for DFS according to the presence of EGFR mutations over all patients. (**c**) Forest plots for risk factors of disease recurrence in stage IB–IIIA disease. (**d**) Kaplan–Meier survival curve for DFS according to the presence of EGFR mutations in stage IB–IIIA disease. NR: not reached; NC: not checked; HR: hazard ratio; CI: confidence interval; DFS: disease-free survival; EGFR: epidermal growth factor receptor.

**Figure 4 cancers-15-05679-f004:**
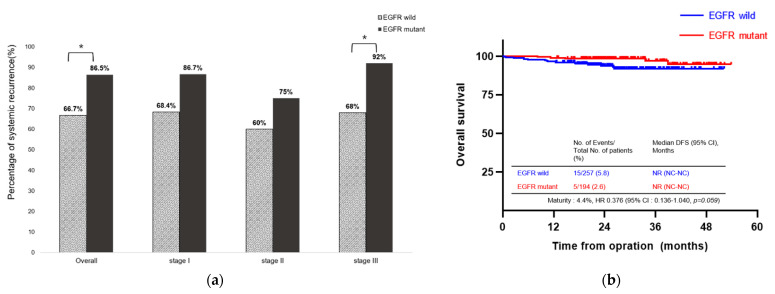
Type of recurrence and overall survival. (**a**) Bar graph showing the proportion of distant metastases among all patients (*p* = 0.016), stage I disease (*p* = 0.257), stage II disease (*p* = 0.652), and stage III disease (*p* = 0.034). (**b**) Kaplan–Meier survival curve for OS according to the presence of EGFR mutations among all patients. * *p* < 0.05. NR: not reached; NC: not checked; HR: hazard ratio; CI: confidence interval; OS: overall survival; EGFR: epidermal growth factor receptor.

**Table 1 cancers-15-05679-t001:** Baseline characteristics.

Characteristic	Total (*n* = 659)	Non-Recurrence (*n* = 529)	Recurrence (*n* = 130)	*p*-Value
Age	65.86 [65.14–66.57]	65.61 [64.82–66.41]	66.86 [65.22–68.50]	0.126
Sex				0.039
Female	260 (39.5)	219 (41.4)	41 (31.5)	
Male	399 (60.5)	310 (58.6)	89 (68.5)	
Smoking status				0.054
Never smoked	327 (49.6)	272 (51.4)	55 (42.3)	
Current smoker	153 (23.2)	124 (23.4)	29 (22.3)	
Ex-smoker	179 (27.2)	133 (25.2)	46 (35.4)	
Second-hand smoking	21 (3.2)	17 (3.2)	4 (3.1)	>0.999
Comorbidity				
HTN	264 (40.1)	209 (39.5)	55 (42.3)	0.560
DM	147 (22.3)	112 (21.2)	35 (26.9)	0.158
Coronary disease	51 (7.7)	37 (7.0)	14 (10.8)	0.149
Vascular disease	2 (0.3)	1 (0.2)	1 (0.8)	0.353
Liver disease				
History of pulmonary tuberculosis	33 (5.0)	26 (4.9)	7 (5.4)	0.826
ILD	14 (2.1)	8 (1.5)	6 (4.6)	0.040
COPD	78 (11)	60 (11.3)	18 (13.8)	0.428
Other malignancy	103 (15.6)	80 (15.1)	23 (17.7)	0.470
Family history	8 (1.2)	8 (1.5)	0 (0.0)	0.223
ECOG PS score				0.007
0	500 (75.9)	414 (78.3)	86 (66.2)	
1	155 (23.5)	113 (21.4)	42 (32.3)	
2	4 (0.6)	2 (0.3)	2 (1.5)	
Pulmonary function (*n* = 650)				
FEV_1_, L	2.41 [2.37–2.45]	2.42 [2.36–2.47]	2.39 [2.30–2.47]	0.825
FVC, L	3.25 [3.14–3.36]	3.26 [3.12–3.39]	3.23 [3.11–3.36]	0.611
DLCO, mL/mmHg/min	17.97 [17.29–18.66]	17.65 [17.27–18.03]	19.29 [16.20–22.37]	0.779
DL_CO_, %	94.16 [90.54–97.78]	93.18 [89.57–96.78]	98.13 [86.99–109.26]	0.608
Serum CEA (*n* = 229)	5.87 [4.39–7.35]	5.59 [4.08–7.10]	6.69 [2.79–10.58]	0.001
Serum proGRP (*n* = 229)	49.79 [45.15–54.42]	50.19 [44.11–56.27]	48.58 [44.60–52.57]	0.137
Serum Cyfra21-1 (*n* = 229)	3.50 [2.66–4.34]	3.45 [2.43–4.47]	3.63 [2.16–5.11]	0.009
Histology				0.243
Adenocarcinoma	491 (74.5)	394 (74.5)	97 (74.6)	
Squamous cell carcinoma	143 (21.7)	118 (22.3)	25 (19.2)	
Other non-small cell carcinoma	25 (3.8)	17 (3.2)	8 (6.2)	
Histologic subtype (*n* = 477)				
Acinar	335 (70.2)	275 (71.8)	60 (63.8)	0.084
Papillary	310 (65.0)	242 (63.2)	68 (72.3)	0.060
Micropapillary	92 (19.3)	59 (15.4)	33 (35.1)	<0.001
Lepidic	136 (28.5)	126 (32.9)	10 (10.6)	<0.001
Solid	112 (23.5)	68 (17.8)	44 (46.8)	<0.001
Cribriform	5 (1.0)	4 (1.0)	1 (1.1)	0.668
Mucinous	36 (7.5)	33 (8.6)	3 (3.2)	0.050
Others	10 (2.1)	9 (2.3)	1 (1.1)	0.695
Pathologic stage (TNM)				<0.001
Stage I	421 (63.9)	377 (71.3)	44 (33.8)	
Stage II	127 (19.3)	100 (18.9)	27 (20.8)	
Stage III	111 (16.8)	52 (9.8)	59 (45.4)	
Driver mutation				
EGFR (*n* = 451)	194 (43.0)	142 (41.2)	52 (49.1)	0.151
ALK (*n* = 453)	26 (5.7)	15 (4.3)	11 (10.6)	0.016
ROS1 (*n* = 441)	7 (1.6)	4 (1.2)	3 (2.9)	0.204
PD-L1 (SP263) (*n* = 479)				0.734
TPS < 1%	295 (61.6)	221 (61.6)	74 (61.6)	
TPS ≥ 1%–< 50%	113 (23.6)	87 (21.2)	26 (21.7)	
TPS ≥ 50%	71 (14.8)	51 (14.2)	20 (16.7)	
Initial therapy				0.121
Operation	645 (97.8)	520 (98.3)	125 (96.1)	
Chemotherapy	5 (0.8)	4 (0.8)	1 (0.8)	
Radiotherapy	1 (0.2)	0 (0.0)	1 (0.8)	
Concurrent chemoradiotherapy	8 (1.2)	5 (0.9)	3 (2.3)	
Operation type				0.954
Wedge resection	13 (2.0)	10 (1.9)	3 (2.3)	
Segmentectomy	37 (5.6)	31 (5.9)	6 (4.6)	
Lobectomy	577 (87.6)	461 (87.1)	116 (89.2)	
Bilobectomy	26 (3.9)	22 (4.2)	4 (3.1)	
Pneumonectomy	6 (0.9)	5 (0.9)	1 (0.8)	
Approach				<0.001
VATS	502 (76.2)	423 (80.0)	79 (60.8)	
Non-VATS	157 (23.8)	106 (20.0)	51 (39.2)	
Tumor location				0.879
Right upper lobe	160 (24.3)	129 (24.4)	31 (23.8)	
Right middle lobe	48 (7.3)	37 (7.0)	11 (8.5)	
Right lower lobe	152 (23.0)	123 (23.3)	29 (22.3)	
Left upper lobe	164 (24.9)	129 (24.4)	35 (26.9)	
Left lower lobe	131 (19.9)	107 (20.2)	24 (18.5)	
Other	4 (0.6)	4 (0.7)	0 (0.0)	
Visceral pleural invasion (*n* = 658)				0.013
Yes	140 (21.3)	102 (19.3)	38 (29.2)	
No	518 (78.7)	426 (80.7)	92 (70.8)	
Lymphatic invasion (*n* = 658)				<0.001
Yes	91 (13.8)	49 (9.3)	42 (32.3)	
No	567 (86.2)	479 (90.7)	88 (67.7)	
Vascular invasion (*n* = 658)				<0.001
Yes	46 (7.0)	24 (4.5)	22 (7.7)	
No	612 (93.0)	504 (95.5)	108 (92.3)	
Neural invasion (*n* = 658)				<0.001
Yes	16 (2.4)	6 (1.1)	10 (7.7)	
No	642 (97.6)	522 (98.9)	120 (92.3)	
Adjuvant therapy (*n* = 243)				0.117
Platinum-based chemotherapy	236 (97.1)	162 (98.2)	74 (94.9)	
Target therapy	1 (0.4)	1 (0.6)	0 (0.0)	
Concurrent chemoradiotherapy	6 (2.5)	2 (1.2)	4 (5.1)	
Survival				<0.001
Alive	572 (86.8)	479 (90.5)	93 (71.5)	
Death	27 (4.1)	10 (1.9)	17 (13.1)	
Censored	60 (9.1)	40 (7.6)	20 (15.4)	

ILD: interstitial lung disease; COPD: chronic obstructive pulmonary disease; ECOG PS score: Eastern Cooperative Oncology Group Performance Score; CEA: carcinoembryonic antigen; proGRP: pro-gastrin releasing peptide; EGFR: epidermal growth factor receptor; ALK: anaplastic lymphoma kinase; ROS1: ROS proto-oncogene 1; PD-L1: programmed death-ligand 1; TPS: tumor proportion score; VATS: video-assisted thoracic surgery. *p*-value from chi-squared test, Fisher’s exact test, or Mann–Whitney U test, as appropriate.

**Table 2 cancers-15-05679-t002:** Prevalence of EGFR subtypes.

EGFR Mutation	Total (*n* = 193)	Non-Recurrence (*n* = 142)	Recurrence (*n* = 51)	*p*-Value
L858R	101 (52.4)	75 (52.8)	26 (50.9)	0.896
L858R only	97 (50.3)	72 (50.7)	25 (49.0)	
L858R + T790M	4 (2.1)	3 (2.1)	1 (1.9)	
Ex19del	79 (40.9)	58 (40.8)	21 (41.2)	
Ex19del only	76 (39.4)	56 (39.4)	20 (39.2)	
Ex19del + T790M	2 (1.0)	1 (0.7)	1 (2.0)	
Ex19del + G719C	1 (0.5)	1 (0.7)	0	
L858R + Ex19del	2 (1.0)	1 (0.7)	1 (2.0)	
Others *	11 (5.7)	8 (5.6)	3 (5.9)	

* Others included 7 patients with E20ins (6 patients in the non-recurrence group and 1 patient in the recurrence group), 2 patients with L861Q (1 patient in the non-recurrence group and 1 patient in the recurrence group), 1 patient with G719C in the non-recurrence group, and 1 patient with G719S in the recurrence group. L858R: exon 21 codon p.Leu858Arg; T790M: Thr790Met; Ex19del: exon 19 deletion; G719C: exon 18 p.Gly719Cys; E20ins: exon 20 insertion; L861Q: exon 21 L861Q.

**Table 3 cancers-15-05679-t003:** Prevalence of recurrence location.

EGFR Status	Wild-Type (*n* = 54)	Mutant (*n* = 52)	Unknown (*n* = 24)	HR [95% CI]	*p*-Value
Lung or regional lymph node	34 (63.0)	26 (50.0)	17 (70.8)	0.588 [0.271–1.277]	0.180
Central nervous system	5 (9.3)	14 (26.9)	4 (16.7)	3.611 [1.195–10.907]	0.023
Bone	17 (31.5)	12 (23.1)	1 (4.2)	0.653 [0.275–1.549]	0.333
Extrathoracic visceral pleura	4 (7.4)	2 (3.8)	1 (4.2)	0.500 [0.088–2.855]	0.435
Pleura	8 (14.8)	14 (26.9)	1 (4.2)	2.118 [0.804–5.583]	0.129
Peritoneum	1 (1.9)	2 (3.8)	0 (0.0)	2.121 [0.186–24.114]	0.545
Head and neck	1 (1.9)	0 (0.0)	0 (0.0)		
Others	5 (9.3)	3 (5.8)	3 (12.5)	0.600 [0.136–2.649]	0.500

## Data Availability

The data presented in this study are available on request from the corresponding author. The data are not publicly available due to institutional data-sharing restrictions.

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
