# Peer review of "Genetic Alterations and Risk Factors for Recurrence in Patients with Non-Small Cell Lung Cancer Who Underwent Complete Surgical Resection"

_cancers, 2023, doi:10.3390/cancers15235679_

Round 1

Reviewer 1 Report

Comments and Suggestions for Authors

Dear Authors,  

This is an interesting topic on an area that needs further research. However, the manuscript overall needs major revisions.  

Specific comments:  

·         Abstract: The study design type, the place/country where the study was conducted, the age of participants, and the OR/HR value should be clearly stated. Abbreviations should be defined on first use (e.g., ROS, ALK). The conclusion does not propose a clear direction for future studies.

·         Keywords list should be expanded.

·         The introduction is too short with no clear rationale on why this study is of importance? The justification of the study should be clear in the last paragraph. Authors also should focus on these points (1) Much progress have been achieved during last decade including combination treatments with chemo+ IO and targeted therapies for mutated patients like EGFR/ALK/ROS1/KRAS pts. Should put more focus on the progress during last years, and despite that to mention that the prognosis still dismal; (2) Authors do not have the strong-enough biological background to review the molecular biology of lung cancer; (3) Smoking increases lung cancer but not all types of lung cancer. For example, some lung cancer patients harboring EGFR mutations do not smoke. This limitation of the link between smoking and lung cancer risk should be highlighted; (4) In Line 75-83, it would be benefit if that authors state the therapeutic role of chemotherapy/irradiation in combination with flavonoid compounds in NSCLC. I would recommend referring to these articles Cancers (Basel). 2019 Oct 15;11(10):1565; Int J Mol Sci. 2022 Jul 18;23(14):7905.

·         Line 89-91: How exactly were the participants recruited? Were they simply asked individually to consent to participation or were there other aspects to the actual recruitment? Were there any peers involved in the process?

·         Line 107-120: Data collection should be described in much more details.

·         Table 3: A column showing a significant P-value should be included.

·         Figure 4 should be clear enough to reader.

·       I believe the discussion is too brief and should be strengthened with more references. Several sentences/statements need references.

·         The scope for future research should be clearly mentioned and discussed in the conclusion.

·         Please include list of abbreviations at the end.

Comments on the Quality of English Language

Moderate English language editing required.

Author Response

[Reviewer 1`s comments]
This is an interesting topic on an area that needs further research. However, the manuscript overall needs major revisions. 

Comment 1: Abstract: The study design type, the place/country where the study was conducted, the age of participants, and the OR/HR value should be clearly stated. Abbreviations should be defined on first use (e.g., ROS, ALK). The conclusion does not propose a clear direction for future studies.

Reply 1: We revised the abstract according to your valuable comments and other reviewer’s opinion.

Changes in the abstract: Definitive surgical resection is the preferred treatment for early-stage non-small-cell lung cancer (NSCLC). Research into genetic alterations, including epidermal growth factor receptor (EGFR) mutation, in early stage NSCLC remains insufficient. Here, we investigated the prevalence of genetic alterations in early-stage NSCLC and the association between EGFR mutation and recurrence after complete resection. Between January 2019 and December 2021, 659 patients with NSCLC who underwent curative surgical resection at a single regional cancer center in Korea were recruited. We retrospectively compared the clinical and pathological data between the recurrence and non-recurrence groups. Multivariate logistic regression was used to predict the risk factors for recurrence. Among the 659 enrolled cases, the median age was 65.86 years old, the most common histology was adenocarcinoma (74.5%), followed by squamous cell carcinoma (21.7%). The prevalence of EGFR mutation was 43% (194/451). Among them, L858R point mutation and exon 19 deletion was 52.3% and 42%, respectively. Anaplastic lymphoma kinase (ALK) rearrangement was found at 5.7% (26/453), and ROS proto-oncogene 1 (ROS1) fusion was found at 1.6% (7/441). The recurrence rate of the entire population was 19.7%. In multivariate analysis, the presence of EGFR mutation (hazard ratio [HR]: 2.698, 95% CI: 1.458–4.993, p = 0.002), stage II (HR: 2.614, 95% CI: 1.29–5.295, p = 0.008) or III (HR: 9.537, 95% CI: 4.825–18.852, p < 0.001) (vs. stage I), and presence of pathologic solid type (HR: 2.598, 95% CI: 1.405–4.803, p = 0.002) were associated with recurrence. Among the recurred group, 86.5% of the patients with EGFR mutation experienced distant recurrence compared to only 66.7% of wild-type (p = 0.016), with no significant difference in median disease-free survival (52.21 months vs. not reached; p = 0.983). In conclusion, adjuvant or neoadjuvant targeted therapy could be considered more actively because EGFR mutation was identified as an independent risk factor for recurrence and associated with systemic recurrence. Further studies on perioperative therapy for other genetic alterations are necessary.

Comment 2. Keywords list should be expanded.

Reply 2: We expanded keyword list as below.

Changes in the keyword: Epidermal growth factor receptor; gene; non-small cell lung cancer; recurrence; surgery

Comment 3. The introduction is too short with no clear rationale on why this study is of importance? The justification of the study should be clear in the last paragraph. Authors also should focus on these points (1) Much progress have been achieved during last decade including combination treatments with chemo+ IO and targeted therapies for mutated patients like EGFR/ALK/ROS1/KRAS pts. Should put more focus on the progress during last years, and despite that to mention that the prognosis still dismal; (2) Authors do not have the strong-enough biological background to review the molecular biology of lung cancer; (3) Smoking increases lung cancer but not all types of lung cancer. For example, some lung cancer patients harboring EGFR mutations do not smoke. This limitation of the link between smoking and lung cancer risk should be highlighted; (4) In Line 75-83, it would be benefit if that authors state the therapeutic role of chemotherapy/irradiation in combination with flavonoid compounds in NSCLC. I would recommend referring to these articles Cancers (Basel). 2019 Oct 15;11(10):1565; Int J Mol Sci. 2022 Jul 18;23(14):7905.

Reply 3: Thank you for your valuable comment. We focused on EGFR mutation in resectable NSCLC and adjuvant targeted therapy. So, there was lack of analysis of PD-L1 or postoperative radiation. We discussed about your comments in introduction part and added more references. We would like to cautiously mention that we will investigate about other perioperative treatment in future study. We modified introduction with reference according to your valuable comments. 

Changes in the text: (Introduction, page 2-3, line 50-112)

In 2020, lung cancer was reported as the leading cause of cancer-related death worldwide and the second most commonly diagnosed cancer.[1] Approximately 48% of lung cancer cases in the U.S. are diagnosed in the early stage.[2,3] Patients with early-stage lung cancer have a better 5-year survival rate compared to those with advanced stage (62.8% vs. 8.2%); therefore, it is crucial to diagnose lung cancer early to ensure the best chance of preventing recurrence.[4]

Definitive surgical resection is the preferred treatment for early stage non-small-cell lung cancer (NSCLC), followed by adjuvant chemotherapy if necessary, depending on the pathological stage.[5] A retrospective analysis revealed that patients with stage I NSCLC who underwent surgical resection had a 5-year survival rate ranging from 40% to 97%.[6] According to a real world study conducted in the U.S., only approximately 40% of patients with NSCLC stage IB-IIIA received adjuvant chemotherapy after surgical resection and 23% of them experienced disease recurrence after 1 year.[7] Therefore, it is crucial to predict recurrence after surgical resection and select patients who are likely to see the greatest benefit from adjuvant therapy. The decision to use adjuvant therapy is mainly based on TNM stage, which has been expanded to include the pathologic findings in the updated guideline. Molecular tests, epidermal growth factor receptor (EGFR) mutation, anaplastic lymphoma kinase (ALK) rearrangement, and programmed death-ligand 1 (PD-L1) expression, are also investigated in patients with early-stage NSCLC.[8]

Smoking increases the risk of some, but not all types of lung cancer. Indeed, some patients with lung cancer harboring EGFR mutations do not smoke. EGFR mutation is the most common driver mutation in advanced NSCLC in East Asia, with a prevalence of approximately 20%–64%.[9] In a large-scale data analysis conducted in China, the EGFR mutation rate in the early stage was 53.6%, which was comparable to the rate observed in the advanced stage (51.4%, p = 0.379). The proportion of EGFR mutations was higher in stage IA than in stages IIB and IIIA. Most EGFR mutations were exon 19 deletion (Ex19del) and exon 21 codon p.Leu858Arg (L858R). In a retrospective multi-center analysis, the presence of EGFR mutation was significantly related to disease recurrence, along with lymphovascular invasion, intrapulmonary metastasis, and lymph node metastasis. And the disease-free survival (DFS) of patients with early stage NSCLC who had EGFR mutation was shorter than that of patients without EGFR mutation even if they shared the same disease stage.[10] Another study showed that the presence of EGFR mutation was a risk factor for distant metastasis in early-stage lung cancer.[11] EGFR mutation, unfavorable factor for recurrence warrants careful consideration. However, studies on the prevalence of EGFR mutations in patients with early-stage NSCLC are limited. Despite this, many studies on EGFR tyrosine kinase inhibitors (EGFR-TKIs) have been expanded to perioperative therapy from palliative therapy in advanced stage.[12-15] as adjuvant therapy have been conducted to lower recurrence and improve prognosis in early-stage NSCLC.

Recently, compared to chemotherapy, adjuvant EGFR-TKIs have demonstrated improved DFS in patients with EGFR-mutated NSCLC. Osimertinib, a third-generation EGFR-TKI, demonstrated a significant improvement in DFS compared to placebo in patients with stage IB–IIIA EGFR-mutated NSCLC who underwent complete resection.15 According to the updated follow-up data of the ADAURA study, the 4-year DFS rate was 73% in the osimertinib group and 38% in the placebo group (overall hazard ratio [HR] for DFS: 0.23; 95% confidence interval [CI]: 0.18 to 0.30).[15] Based on these successful results, osimertinib is recommended as an adjuvant treatment for resected EGFR-mutant stage IB–IIIA NSCLC.[16,17] A previous systemic review revealed that adjuvant EGFR-TKI prolonged DFS in patients with EGFR mutation, regardless of the mutation subtype, but could not prolong overall survival (OS).[18] Given these suboptimal results, platinum-doublet chemotherapy is still preferred as an adjuvant therapy in NSCLC.[8] Additionally, it has been suggested that natural polyphenols have a potential contribution to lung cancer treatment.[19]

Despite much progress during last decade, including targeted therapy and immunotherapy in advanced stage, the prognosis of NSCLC remains dismal. Therefore, more information is needed on patients with early-stage disease, especially in Asian populations, which tend to have a large number of non-smokers. In this study, we investigated the prevalence and importance of genetic alteration in patients with early-stage NSCLC who underwent definitive surgery in a single regional cancer center of Korea.

Comment 4. How exactly were the participants recruited? Were they simply asked individually to consent to participation or were there other aspects to the actual recruitment? Were there any peers involved in the process?

Reply 4: We retrospectively reviewed electronic medical record who received surgical resection with NSCLC from Jan 1st 2019 to Dec 31st 2021. In the review of medical records, the two pulmonologists participated. We modified the word ‘recruited’ to ‘collected’ and add the description of data analysis method.

Changes in the text: (Materials and Methods, page 3, line 115-118)

We collected data on 1,095 patients who underwent surgical resection at the Lung Cancer Center, Chonnam National University Hwasun Hospital (CNUHH) from January 1, 2019 to December 31, 2021 and retrospectively reviewed their electronic medical records (Fig. 1).

Comment 5. Data collection should be described in much more details

Reply 5: We added details about data collection.

Changes in the text: (Materials and Methods, page 4, line 134-148)

We collected clinical and pathological information of 659 patients. The collected data included sex, age, history of smoking or second-hand smoking, comorbidities (e.g., hypertension, diabetes mellitus, coronary disease, peripheral vascular disease, liver disease, history of pulmonary tuberculosis, interstitial lung disease, chronic obstructive pulmonary disease, or other malignancy), family history of lung cancer, Eastern Cooperative Oncology Group Performance Score (ECOG PS), parameters of pulmonary function test (e.g., FEV1, FVC, and DLCO), serum carcinoembryonic antigen (CEA), serum pro-gastrin releasing peptide (proGRP), serum Cyfra21-1, histologic type, histologic subtype, pathologic stage, presence of EGFR mutation, anaplastic lymphoma kinase (ALK) re-arrangement, ROS proto-oncogene 1 (ROS1) fusion, programmed death-ligand 1 (PD-L1) expression, initial therapy, operation type, surgical approach type, date of surgery, tumor location, presence of visceral pleural invasion, lymphatic invasion, vascular and neural invasion, implementation of adjuvant therapy, recurrence state, date of recurrence, survival state, and date of last follow-up. The recurrence date was set the date on which the treatment for recurrence began.

Comment 6. A column showing a significant P-value should be included in table 3.

Reply 6: We showed p-value of each location of distant metastasis in table 3. Also, we detected error of the number of patients whose EGFR mutation were unknown, so we corrected the number of them.

Changes in the text: (Results, page 11, line 307)

Comment 7. Figure 4 should be clear enough to reader.

Reply 7: We changed the figure 3b, 3d and 4b to colored version.

Changes in the text: We submitted a revised figure 3 and figure 4.

Fig. 3

(a)

EGFR wild

EGFR mutant

 (b)

(c)

 (d)

Fig. 4

(a)

(b)

Comment 8. Discussion is too brief and should be strengthened with more references.

Reply 8: Thank you for your valuable comment. We investigated about association with EGFR and high risk of recurrence and results of previous studies about adjuvant therapy in NSCLC after surgical resection. We modified the text of discussion and reinforced the references about the prevalence of EGFR mutation,

Changes in the text: (Discussion, page 12, line 323-326) We found that approximately two-thirds of all patients were tested for genetic alterations, among whom, 43% had EGFR mutations. The prevalence of EGFR mutations in this study was similar with that of Asians in previous studies.[27,28]

(Discussion, page 12, line 327-333) Various prognostic factors have been found to be associated with disease recurrence in early-stage NSCLC after complete resection, including age, histologic pattern, lymphovascular invasion, invasive size, pathologic stage, and genetic alterations. The significance of EGFR mutation as a prognostic factor for early-stage NSCLC has been reported along with various factors such as age, histologic pattern, pathologic stage, and presence of lymphovascular invasion.[29-33]

(Discussion, page 11, line 336-343) However, the difference in DFS between EGFR-mutant and wild-type NSCLC was not statistically significant. In our study, we found that the presence of EGFR mutation, pathologic subtype (presence of pathologic solid subtype, and advanced stage were associated with recurrence in all patients. In addition, advanced stage and presence of pathologic subtype pathologic acinar subtype or mucinous subtype were associated with recurrence in patients with stage IB–IIIA disease. We have discovered that EGFR mutation could be an impactive factor in recurrence, which is consistent with the findings of previous studies.

(Discussion, page 12, line 344-356) EGFR mutation has been reported to be harbored more frequently in patients with a lepidic component than those without.[34,35] A lepidic component is a favorable prognostic factor and usually has a part-solid appearance in computed tomography.[36,37] The volume and diameter of ground glass opacity have been shown to be correlated with EGFR mutation, while the presence of EGFR mutation has been shown to be correlated with the growth of ground glass nodule.[38,39] Moreover, a previous study revealed that EGFR mutation, especially L858R mutation, increased the cell invasion ability in lung adenocarcinoma.[40] Additionally, it is theorized that circulating tumor cells increase in the draining pulmonary vein during surgical resection, which is particularly significant in patients with lymphatic invasion.[41] Taken together, these findings infer that EGFR mutation is associated with recurrence after surgical resection due to the high invasiveness.

(Discussion, page 12, line 357-358) Although the EGFR mutation did not impact DFS and OS in this study, it was associated with a higher risk of recurrence and the type of metastasis.  

(Discussion, page 12, line 362-375) Among sites of distant recurrence, the brain is the most common site. It has long been known that the patients with brain metastases have particularly unfavorable prognoses.[44] Despite of the poor prognosis of patients who experience recurrence after surgical resection, less than 50% of patients who underwent definitive surgery for early-stage NSCLC received adjuvant therapy.[7,45] Furthermore, the proportion of patients with EGFR-positive NSCLC who received adjuvant chemotherapy, not a targeted therapy, after definitive resection is still high.[46,47] In the real world, more than 95% of patients with EGFR-positive NSCLC who received adjuvant therapy received platinum-based chemotherapy. The situation was similar in our center. We discovered that 23% of patients received adjuvant therapy after curative resection, among whom, only one patient received target therapy. However, platinum-based adjuvant chemotherapy has been shown to be ineffective in preventing CNS recurrence.[48]

(Discussion, page 13, line 376-378) The recently updated guideline recommend investigation of EGFR mutation status and ALK rearrangement for adjuvant therapy in resectable NSCLC.[8]

(Discussion, page 13, line 395-400) CT-guided core needle biopsy, which is a commonly used biopsy modality, has a sensitivity of 90.07% and accuracy of 98.87%, with an AUC of 0.952. Additionally, it is rare for serious complications to occur after CT-guided core needle biopsy.[50] Along with this safe and accurate diagnostic tool, if the genetic alterations can be found with a more sensitive method such as liquid biopsy, the prognosis of lung cancer might be improved with neoadjuvant targeted therapy.

Comment 9. The scope for future research should be clearly mentioned and discussed in the conclusion

Reply 9: We thought more about future research and direction which we should attempt.

Changed in the text: (Conclusion, page 13-14, line 411-423) In the present study, we showed that the prevalence of EGFR mutation, ALK rearrangement, and ROS1 fusion was 43.0%, 5.7%, and 1.6% in patients with NSCLC who underwent complete resection, respectively. EGFR mutation was an independent risk factor for recurrence along with stage II/III and solid pathologic subtype and was associated with a high rate of distant metastasis and a higher risk of CNS recurrence compared to wild-type EGFR. Among patients with recurrent malignancy, the rate of distant metastasis was higher in patients with EGFR mutations than in patients without EGFR mutations, and EGFR mu-tations were associated with CNS recurrence. Therefore, adjuvant or neoadjuvant targeted therapy could be considered more actively in patients with EGFR mutation who have undergone curative surgery. Further studies on perioperative therapy for other genetic alterations are also needed, and it is important to establish more sensitive methods, such as liquid biopsy, to predict recurrence and provide optimal therapy.

Comment 10. Include list of abbreviations at the end.

Reply 10: Thank you for your comment. We described list of abbreviation at the end.

Changes in the text: (page 14, line 446-458) Abbreviations: ALK: Anaplastic lymphoma kinase, cDNA: Circulating DNA, CEA: Carcinoembryonic antigen, CI: Confidence intervals, CNS: Central nervous system, COPD: Chronic obstructive pulmonary disease, DM: Diabetes mellitus, ECOG PS score: Eastern Cooperative Oncology Group Performance Score, DFS: Disease-free survival, E20ins: Exon 20 insertions, EGFR: Epidermal growth factor receptor, EGFR-TKI: Epidermal growth factor receptor-tyrosine kinase inhibitor, Ex19del: Exon 19 deletion, FFPE: Formalin-fixed paraffin-embedded, G719C: Exon 18 p.Gly719Cys, HTN: Hypertension, HR: Hazard ratio, ILD: Interstitial lung disease, L858R: Exon 21 codon p.Leu858Arg, L861Q: Exon 21 L861Q, MLND: Mediastinal lymph node dissection, NC: Not checked, NR: Not reached, NSCLC: Non-small cell lung carcinoma, O&C: Open and closure, OS: Overall survival, PCR: Polymerase chain reaction, PD-L1: Programmed death-ligand 1, proGRP: Pro-gastrin releasing peptide, ROS1: ROS proto-oncogene 1, T790M: Thr790Met, TPS: Tumor proportion score, VATS: Video-assisted thoracic surgery

Manuscript ID: cancers-2724441

Title: Genetic alterations and risk factors for recurrence in patients with non-small cell lung cancer who underwent complete surgical resection

Dear the Editors of Cancers,

Thank you for giving us the opportunity to submit a revised manuscript again. We appreciate the time and effort that you and the reviewers dedicated to providing feedback on our manuscript and are grateful for the insightful comments to improve our paper.

We have incorporated an additional suggestion made by the reviewers. These changes are highlighted as point-by-point response to the reviewer’s comment. In addition, language corrections by professional English editing service are shown by MS word track changes.

We hope that all changes we have made meet with your approval and look forward to your response.

Sincerely yours,

In-Jae Oh, on behalf of all the authors.

Department of Internal Medicine, Chonnam National University Hwasun Hospital,

322 Seoyang-ro, Hwasun, Jeonnam 58128, Republic of Korea

Tel.: 061-379-7617

Fax: 061-379-7619

E-mail: droij@jnu.ac.kr

[Reviewer 1`s comments]
This is an interesting topic on an area that needs further research. However, the manuscript overall needs major revisions. 

Comment 1: Abstract: The study design type, the place/country where the study was conducted, the age of participants, and the OR/HR value should be clearly stated. Abbreviations should be defined on first use (e.g., ROS, ALK). The conclusion does not propose a clear direction for future studies.

Reply 1: We revised the abstract according to your valuable comments and other reviewer’s opinion.

Changes in the abstract: Definitive surgical resection is the preferred treatment for early-stage non-small-cell lung cancer (NSCLC). Research into genetic alterations, including epidermal growth factor receptor (EGFR) mutation, in early stage NSCLC remains insufficient. Here, we investigated the prevalence of genetic alterations in early-stage NSCLC and the association between EGFR mutation and recurrence after complete resection. Between January 2019 and December 2021, 659 patients with NSCLC who underwent curative surgical resection at a single regional cancer center in Korea were recruited. We retrospectively compared the clinical and pathological data between the recurrence and non-recurrence groups. Multivariate logistic regression was used to predict the risk factors for recurrence. Among the 659 enrolled cases, the median age was 65.86 years old, the most common histology was adenocarcinoma (74.5%), followed by squamous cell carcinoma (21.7%). The prevalence of EGFR mutation was 43% (194/451). Among them, L858R point mutation and exon 19 deletion was 52.3% and 42%, respectively. Anaplastic lymphoma kinase (ALK) rearrangement was found at 5.7% (26/453), and ROS proto-oncogene 1 (ROS1) fusion was found at 1.6% (7/441). The recurrence rate of the entire population was 19.7%. In multivariate analysis, the presence of EGFR mutation (hazard ratio [HR]: 2.698, 95% CI: 1.458–4.993, p = 0.002), stage II (HR: 2.614, 95% CI: 1.29–5.295, p = 0.008) or III (HR: 9.537, 95% CI: 4.825–18.852, p < 0.001) (vs. stage I), and presence of pathologic solid type (HR: 2.598, 95% CI: 1.405–4.803, p = 0.002) were associated with recurrence. Among the recurred group, 86.5% of the patients with EGFR mutation experienced distant recurrence compared to only 66.7% of wild-type (p = 0.016), with no significant difference in median disease-free survival (52.21 months vs. not reached; p = 0.983). In conclusion, adjuvant or neoadjuvant targeted therapy could be considered more actively because EGFR mutation was identified as an independent risk factor for recurrence and associated with systemic recurrence. Further studies on perioperative therapy for other genetic alterations are necessary.

Comment 2. Keywords list should be expanded.

Reply 2: We expanded keyword list as below.

Changes in the keyword: Epidermal growth factor receptor; gene; non-small cell lung cancer; recurrence; surgery

Comment 3. The introduction is too short with no clear rationale on why this study is of importance? The justification of the study should be clear in the last paragraph. Authors also should focus on these points (1) Much progress have been achieved during last decade including combination treatments with chemo+ IO and targeted therapies for mutated patients like EGFR/ALK/ROS1/KRAS pts. Should put more focus on the progress during last years, and despite that to mention that the prognosis still dismal; (2) Authors do not have the strong-enough biological background to review the molecular biology of lung cancer; (3) Smoking increases lung cancer but not all types of lung cancer. For example, some lung cancer patients harboring EGFR mutations do not smoke. This limitation of the link between smoking and lung cancer risk should be highlighted; (4) In Line 75-83, it would be benefit if that authors state the therapeutic role of chemotherapy/irradiation in combination with flavonoid compounds in NSCLC. I would recommend referring to these articles Cancers (Basel). 2019 Oct 15;11(10):1565; Int J Mol Sci. 2022 Jul 18;23(14):7905.

Reply 3: Thank you for your valuable comment. We focused on EGFR mutation in resectable NSCLC and adjuvant targeted therapy. So, there was lack of analysis of PD-L1 or postoperative radiation. We discussed about your comments in introduction part and added more references. We would like to cautiously mention that we will investigate about other perioperative treatment in future study. We modified introduction with reference according to your valuable comments. 

Changes in the text: (Introduction, page 2-3, line 50-112)

In 2020, lung cancer was reported as the leading cause of cancer-related death worldwide and the second most commonly diagnosed cancer.[1] Approximately 48% of lung cancer cases in the U.S. are diagnosed in the early stage.[2,3] Patients with early-stage lung cancer have a better 5-year survival rate compared to those with advanced stage (62.8% vs. 8.2%); therefore, it is crucial to diagnose lung cancer early to ensure the best chance of preventing recurrence.[4]

Definitive surgical resection is the preferred treatment for early stage non-small-cell lung cancer (NSCLC), followed by adjuvant chemotherapy if necessary, depending on the pathological stage.[5] A retrospective analysis revealed that patients with stage I NSCLC who underwent surgical resection had a 5-year survival rate ranging from 40% to 97%.[6] According to a real world study conducted in the U.S., only approximately 40% of patients with NSCLC stage IB-IIIA received adjuvant chemotherapy after surgical resection and 23% of them experienced disease recurrence after 1 year.[7] Therefore, it is crucial to predict recurrence after surgical resection and select patients who are likely to see the greatest benefit from adjuvant therapy. The decision to use adjuvant therapy is mainly based on TNM stage, which has been expanded to include the pathologic findings in the updated guideline. Molecular tests, epidermal growth factor receptor (EGFR) mutation, anaplastic lymphoma kinase (ALK) rearrangement, and programmed death-ligand 1 (PD-L1) expression, are also investigated in patients with early-stage NSCLC.[8]

Smoking increases the risk of some, but not all types of lung cancer. Indeed, some patients with lung cancer harboring EGFR mutations do not smoke. EGFR mutation is the most common driver mutation in advanced NSCLC in East Asia, with a prevalence of approximately 20%–64%.[9] In a large-scale data analysis conducted in China, the EGFR mutation rate in the early stage was 53.6%, which was comparable to the rate observed in the advanced stage (51.4%, p = 0.379). The proportion of EGFR mutations was higher in stage IA than in stages IIB and IIIA. Most EGFR mutations were exon 19 deletion (Ex19del) and exon 21 codon p.Leu858Arg (L858R). In a retrospective multi-center analysis, the presence of EGFR mutation was significantly related to disease recurrence, along with lymphovascular invasion, intrapulmonary metastasis, and lymph node metastasis. And the disease-free survival (DFS) of patients with early stage NSCLC who had EGFR mutation was shorter than that of patients without EGFR mutation even if they shared the same disease stage.[10] Another study showed that the presence of EGFR mutation was a risk factor for distant metastasis in early-stage lung cancer.[11] EGFR mutation, unfavorable factor for recurrence warrants careful consideration. However, studies on the prevalence of EGFR mutations in patients with early-stage NSCLC are limited. Despite this, many studies on EGFR tyrosine kinase inhibitors (EGFR-TKIs) have been expanded to perioperative therapy from palliative therapy in advanced stage.[12-15] as adjuvant therapy have been conducted to lower recurrence and improve prognosis in early-stage NSCLC.

Recently, compared to chemotherapy, adjuvant EGFR-TKIs have demonstrated improved DFS in patients with EGFR-mutated NSCLC. Osimertinib, a third-generation EGFR-TKI, demonstrated a significant improvement in DFS compared to placebo in patients with stage IB–IIIA EGFR-mutated NSCLC who underwent complete resection.15 According to the updated follow-up data of the ADAURA study, the 4-year DFS rate was 73% in the osimertinib group and 38% in the placebo group (overall hazard ratio [HR] for DFS: 0.23; 95% confidence interval [CI]: 0.18 to 0.30).[15] Based on these successful results, osimertinib is recommended as an adjuvant treatment for resected EGFR-mutant stage IB–IIIA NSCLC.[16,17] A previous systemic review revealed that adjuvant EGFR-TKI prolonged DFS in patients with EGFR mutation, regardless of the mutation subtype, but could not prolong overall survival (OS).[18] Given these suboptimal results, platinum-doublet chemotherapy is still preferred as an adjuvant therapy in NSCLC.[8] Additionally, it has been suggested that natural polyphenols have a potential contribution to lung cancer treatment.[19]

Despite much progress during last decade, including targeted therapy and immunotherapy in advanced stage, the prognosis of NSCLC remains dismal. Therefore, more information is needed on patients with early-stage disease, especially in Asian populations, which tend to have a large number of non-smokers. In this study, we investigated the prevalence and importance of genetic alteration in patients with early-stage NSCLC who underwent definitive surgery in a single regional cancer center of Korea.

Comment 4. How exactly were the participants recruited? Were they simply asked individually to consent to participation or were there other aspects to the actual recruitment? Were there any peers involved in the process?

Reply 4: We retrospectively reviewed electronic medical record who received surgical resection with NSCLC from Jan 1st 2019 to Dec 31st 2021. In the review of medical records, the two pulmonologists participated. We modified the word ‘recruited’ to ‘collected’ and add the description of data analysis method.

Changes in the text: (Materials and Methods, page 3, line 115-118)

We collected data on 1,095 patients who underwent surgical resection at the Lung Cancer Center, Chonnam National University Hwasun Hospital (CNUHH) from January 1, 2019 to December 31, 2021 and retrospectively reviewed their electronic medical records (Fig. 1).

Comment 5. Data collection should be described in much more details

Reply 5: We added details about data collection.

Changes in the text: (Materials and Methods, page 4, line 134-148)

We collected clinical and pathological information of 659 patients. The collected data included sex, age, history of smoking or second-hand smoking, comorbidities (e.g., hypertension, diabetes mellitus, coronary disease, peripheral vascular disease, liver disease, history of pulmonary tuberculosis, interstitial lung disease, chronic obstructive pulmonary disease, or other malignancy), family history of lung cancer, Eastern Cooperative Oncology Group Performance Score (ECOG PS), parameters of pulmonary function test (e.g., FEV1, FVC, and DLCO), serum carcinoembryonic antigen (CEA), serum pro-gastrin releasing peptide (proGRP), serum Cyfra21-1, histologic type, histologic subtype, pathologic stage, presence of EGFR mutation, anaplastic lymphoma kinase (ALK) re-arrangement, ROS proto-oncogene 1 (ROS1) fusion, programmed death-ligand 1 (PD-L1) expression, initial therapy, operation type, surgical approach type, date of surgery, tumor location, presence of visceral pleural invasion, lymphatic invasion, vascular and neural invasion, implementation of adjuvant therapy, recurrence state, date of recurrence, survival state, and date of last follow-up. The recurrence date was set the date on which the treatment for recurrence began.

Comment 6. A column showing a significant P-value should be included in table 3.

Reply 6: We showed p-value of each location of distant metastasis in table 3. Also, we detected error of the number of patients whose EGFR mutation were unknown, so we corrected the number of them.

Changes in the text: (Results, page 11, line 307)

Comment 7. Figure 4 should be clear enough to reader.

Reply 7: We changed the figure 3b, 3d and 4b to colored version.

Changes in the text: We submitted a revised figure 3 and figure 4.

Fig. 3

(a)

EGFR wild

EGFR mutant

 (b)

(c)

 (d)

Fig. 4

(a)

(b)

Comment 8. Discussion is too brief and should be strengthened with more references.

Reply 8: Thank you for your valuable comment. We investigated about association with EGFR and high risk of recurrence and results of previous studies about adjuvant therapy in NSCLC after surgical resection. We modified the text of discussion and reinforced the references about the prevalence of EGFR mutation,

Changes in the text: (Discussion, page 12, line 323-326) We found that approximately two-thirds of all patients were tested for genetic alterations, among whom, 43% had EGFR mutations. The prevalence of EGFR mutations in this study was similar with that of Asians in previous studies.[27,28]

(Discussion, page 12, line 327-333) Various prognostic factors have been found to be associated with disease recurrence in early-stage NSCLC after complete resection, including age, histologic pattern, lymphovascular invasion, invasive size, pathologic stage, and genetic alterations. The significance of EGFR mutation as a prognostic factor for early-stage NSCLC has been reported along with various factors such as age, histologic pattern, pathologic stage, and presence of lymphovascular invasion.[29-33]

(Discussion, page 11, line 336-343) However, the difference in DFS between EGFR-mutant and wild-type NSCLC was not statistically significant. In our study, we found that the presence of EGFR mutation, pathologic subtype (presence of pathologic solid subtype, and advanced stage were associated with recurrence in all patients. In addition, advanced stage and presence of pathologic subtype pathologic acinar subtype or mucinous subtype were associated with recurrence in patients with stage IB–IIIA disease. We have discovered that EGFR mutation could be an impactive factor in recurrence, which is consistent with the findings of previous studies.

(Discussion, page 12, line 344-356) EGFR mutation has been reported to be harbored more frequently in patients with a lepidic component than those without.[34,35] A lepidic component is a favorable prognostic factor and usually has a part-solid appearance in computed tomography.[36,37] The volume and diameter of ground glass opacity have been shown to be correlated with EGFR mutation, while the presence of EGFR mutation has been shown to be correlated with the growth of ground glass nodule.[38,39] Moreover, a previous study revealed that EGFR mutation, especially L858R mutation, increased the cell invasion ability in lung adenocarcinoma.[40] Additionally, it is theorized that circulating tumor cells increase in the draining pulmonary vein during surgical resection, which is particularly significant in patients with lymphatic invasion.[41] Taken together, these findings infer that EGFR mutation is associated with recurrence after surgical resection due to the high invasiveness.

(Discussion, page 12, line 357-358) Although the EGFR mutation did not impact DFS and OS in this study, it was associated with a higher risk of recurrence and the type of metastasis.  

(Discussion, page 12, line 362-375) Among sites of distant recurrence, the brain is the most common site. It has long been known that the patients with brain metastases have particularly unfavorable prognoses.[44] Despite of the poor prognosis of patients who experience recurrence after surgical resection, less than 50% of patients who underwent definitive surgery for early-stage NSCLC received adjuvant therapy.[7,45] Furthermore, the proportion of patients with EGFR-positive NSCLC who received adjuvant chemotherapy, not a targeted therapy, after definitive resection is still high.[46,47] In the real world, more than 95% of patients with EGFR-positive NSCLC who received adjuvant therapy received platinum-based chemotherapy. The situation was similar in our center. We discovered that 23% of patients received adjuvant therapy after curative resection, among whom, only one patient received target therapy. However, platinum-based adjuvant chemotherapy has been shown to be ineffective in preventing CNS recurrence.[48]

(Discussion, page 13, line 376-378) The recently updated guideline recommend investigation of EGFR mutation status and ALK rearrangement for adjuvant therapy in resectable NSCLC.[8]

(Discussion, page 13, line 395-400) CT-guided core needle biopsy, which is a commonly used biopsy modality, has a sensitivity of 90.07% and accuracy of 98.87%, with an AUC of 0.952. Additionally, it is rare for serious complications to occur after CT-guided core needle biopsy.[50] Along with this safe and accurate diagnostic tool, if the genetic alterations can be found with a more sensitive method such as liquid biopsy, the prognosis of lung cancer might be improved with neoadjuvant targeted therapy.

Comment 9. The scope for future research should be clearly mentioned and discussed in the conclusion

Reply 9: We thought more about future research and direction which we should attempt.

Changed in the text: (Conclusion, page 13-14, line 411-423) In the present study, we showed that the prevalence of EGFR mutation, ALK rearrangement, and ROS1 fusion was 43.0%, 5.7%, and 1.6% in patients with NSCLC who underwent complete resection, respectively. EGFR mutation was an independent risk factor for recurrence along with stage II/III and solid pathologic subtype and was associated with a high rate of distant metastasis and a higher risk of CNS recurrence compared to wild-type EGFR. Among patients with recurrent malignancy, the rate of distant metastasis was higher in patients with EGFR mutations than in patients without EGFR mutations, and EGFR mu-tations were associated with CNS recurrence. Therefore, adjuvant or neoadjuvant targeted therapy could be considered more actively in patients with EGFR mutation who have undergone curative surgery. Further studies on perioperative therapy for other genetic alterations are also needed, and it is important to establish more sensitive methods, such as liquid biopsy, to predict recurrence and provide optimal therapy.

Comment 10. Include list of abbreviations at the end.

Reply 10: Thank you for your comment. We described list of abbreviation at the end.

Changes in the text: (page 14, line 446-458) Abbreviations: ALK: Anaplastic lymphoma kinase, cDNA: Circulating DNA, CEA: Carcinoembryonic antigen, CI: Confidence intervals, CNS: Central nervous system, COPD: Chronic obstructive pulmonary disease, DM: Diabetes mellitus, ECOG PS score: Eastern Cooperative Oncology Group Performance Score, DFS: Disease-free survival, E20ins: Exon 20 insertions, EGFR: Epidermal growth factor receptor, EGFR-TKI: Epidermal growth factor receptor-tyrosine kinase inhibitor, Ex19del: Exon 19 deletion, FFPE: Formalin-fixed paraffin-embedded, G719C: Exon 18 p.Gly719Cys, HTN: Hypertension, HR: Hazard ratio, ILD: Interstitial lung disease, L858R: Exon 21 codon p.Leu858Arg, L861Q: Exon 21 L861Q, MLND: Mediastinal lymph node dissection, NC: Not checked, NR: Not reached, NSCLC: Non-small cell lung carcinoma, O&C: Open and closure, OS: Overall survival, PCR: Polymerase chain reaction, PD-L1: Programmed death-ligand 1, proGRP: Pro-gastrin releasing peptide, ROS1: ROS proto-oncogene 1, T790M: Thr790Met, TPS: Tumor proportion score, VATS: Video-assisted thoracic surgery

Reviewer 2 Report

Comments and Suggestions for Authors

The topic is interesting. The manuscript quite well written. I suggest some changes to improve the manuscript:

1) Abstract. Definitive surgical resection is the preferred treatment for early-stage non-small-cell lung cancer (NSCLC). Research into genetic alterations, including epidermal growth factor receptor (EGFR) mutation, in early stage NSCLC remains insufficient. Here, we investigated the prevalence of genetic alterations in early-stage NSCLC and the association between EGFR mutation and recurrence after complete resection. Between January 2019 and December 2021, 659 patients with NSCLC who underwent curative surgical resection at a single regional cancer center were recruited. We compared the clinical and pathological data between the recurrence and non-recurrence groups. Multivariate logistic regression was used to predict the risk factors for recurrence. Among the 659 enrolled cases, the most common histology was adenocarcinoma (74.5%), followed by squamous cell carcinoma (21.7%). The prevalence of EGFR mutation was 43% (194/451). Among them, L858R point mutation and exon 19 deletion was 52.3% and 42%, respectively. ALK rearrangement was found at 5.7% (26/453), and ROS1 fusion was found at 1.6% (7/441). The recurrence rate of the entire population was 19.7%. In multivariate analysis, the presence of EGFR mutation, stage II or III (vs. stage I), and pathologic subtype (presence of solid type) were associated with recurrence. Among the recurred group, 86.5% of the patients with EGFR mutation experienced distant recurrence compared to only 66.7% of wild-type (p = 0.016), with no significant difference in median disease-free survival (p = 0.983). In conclusion, the prevalence of EGFR mutation, ALK rearrangement, and ROS1 fusion was 43.0%, 5.7%, and 1.6%, respectively in patients with early-stage NSCLC who underwent curative resection. Along with stage II/III and solid pathologic subtype, EGFR mutation was an independent risk factor for recurrence. In the recurrence group, the rate of distant metastasis was higher in patients with EGFR mutation than in those with wild-type. Abstract: It might be beneficial to include a sentence in the abstract that briefly summarizes the key findings of the study. This can provide readers with a quick overview of the research. 

2) 1. Introduction 45 In 2020, lung cancer was reported as the leading cause of death worldwide and the 46 second most commonly diagnosed cancer.[1] Approximately 48% of lung cancer cases in 47 the U.S. are diagnosed in the early stage.[2, 3] Patients with early-stage lung cancer have 48 a better 5-year survival rate compared to those with advanced stage (62.8% vs. 8.2%); 49 therefore, it is crucial to diagnose lung cancer early to ensure the best chance of preventing 50 recurrence.[4] 51 Definitive surgical resection is the preferred treatment for early stage non-small-cell 52 lung cancer (NSCLC), followed by adjuvant chemotherapy if necessary, depending on the 53 pathological stage.[5] A retrospective analysis revealed that patients with stage I NSCLC 54 who underwent surgical resection had a 5-year survival rate ranging from 40% to 97%.[6] 55 In the real world, it has been reported that 23% of patients with early stage NSCLC who 56 underwent surgical resection experienced disease recurrence after 1 year. Furthermore, 57 the 5-year disease-free survival (DFS) rate was 29.3%, which declined as the disease pro- 58 gressed.[7].  I suggest that you include some information in order to complete the manuscript. Below you can find some works that could give useful ideas in expanding this part:

1- Accuracy of CT-Guided Core-Needle Biopsy in Diagnosis of Thoracic Lesions Suspicious for Primitive Malignancy of the Lung: A Five-Year Retrospective Analysis. Tomography. 2022;8(6):2828-2838. doi: 10.3390/tomography8060236. 

2- CT-guided percutaneous core needle biopsy for small (≤20 mm) pulmonary lesions. Clin Radiol. 2013 Jan;68(1):e43-8. doi: 10.1016/j.crad.2012.09.008.

3- CCN Guidelines® Insights: Non-Small Cell Lung Cancer, Version 2.2023. J Natl Compr Canc Netw. 2023 Apr;21(4):340-350. doi: 10.6004/jnccn.2023.0020. 

3) In this study, we investigated the prevalence of genetic alteration in patients with 84 early-stage NSCLC who underwent definitive surgery in a single center, as well as the 85 association between EGFR mutation and recurrence after curative surgery. Please, improve the description of study aim.

4) 5. Conclusion 357 In the present study, we showed that the prevalence of EGFR mutation, ALK rear- 358 rangement, and ROS1 fusion was 43.0%, 5.7%, and 1.6% in patients with NSCLC who 359 underwent complete resection, respectively. EGFR mutation was an independent risk fac- 360 tor for recurrence along with stage II/III and solid pathologic subtype. Among patients 361 with recurrent malignancy, the rate of distant metastasis was higher in patients with EGFR 362 mutations than in patients without EGFR mutations, and EGFR mutations were associated 363 with CNS recurrence. With regard to the promising results of EGFR-TKI as adjuvant ther- 364 apy, it is advisable to identify the histological characteristics of patients with NSCLC who 365 underwent curative surgery and actively consider adjuvant targeted therapy. Please, underline the novelty of the study and ameliorate the description of possible clinical implications. 

Comments on the Quality of English Language

 Minor changes of English language are required

Author Response

[Reviewer 2`s comments]
The topic is interesting. The manuscript quite well written. I suggest some changes to improve the manuscript:

Comment 1. Abstract: It might be beneficial to include a sentence in the abstract that briefly summarizes the key findings of the study. This can provide readers with a quick overview of the research.

Reply 1: Thank you for your kind reply. We summarized key results of our study, and described in abstract according to 3 reviewers’ comments.

Changes in the abstract: Definitive surgical resection is the preferred treatment for early-stage non-small-cell lung cancer (NSCLC). Research into genetic alterations, including epidermal growth factor receptor (EGFR) mutation, in early stage NSCLC remains insufficient. Here, we investigated the prevalence of genetic alterations in early-stage NSCLC and the association between EGFR mutation and recurrence after complete resection. Between January 2019 and December 2021, 659 patients with NSCLC who underwent curative surgical resection at a single regional cancer center in Korea were recruited. We retrospectively compared the clinical and pathological data between the recurrence and non-recurrence groups. Multivariate logistic regression was used to predict the risk factors for recurrence. Among the 659 enrolled cases, the median age was 65.86 years old, the most common histology was adenocarcinoma (74.5%), followed by squamous cell carcinoma (21.7%). The prevalence of EGFR mutation was 43% (194/451). Among them, L858R point mutation and exon 19 deletion was 52.3% and 42%, respectively. Anaplastic lymphoma kinase (ALK) rearrangement was found at 5.7% (26/453), and ROS proto-oncogene 1 (ROS1) fusion was found at 1.6% (7/441). The recurrence rate of the entire population was 19.7%. In multivariate analysis, the presence of EGFR mutation (hazard ratio [HR]: 2.698, 95% CI: 1.458–4.993, p = 0.002), stage II (HR: 2.614, 95% CI: 1.29–5.295, p = 0.008) or III (HR: 9.537, 95% CI: 4.825–18.852, p < 0.001) (vs. stage I), and presence of pathologic solid type (HR: 2.598, 95% CI: 1.405–4.803, p = 0.002) were associated with recurrence. Among the recurred group, 86.5% of the patients with EGFR mutation experienced distant recurrence compared to only 66.7% of wild-type (p = 0.016), with no significant difference in median disease-free survival (52.21 months vs. not reached; p = 0.983). In conclusion, adjuvant or neoadjuvant targeted therapy could be considered more actively because EGFR mutation was identified as an independent risk factor for recurrence and associated with systemic recurrence. Further studies on perioperative therapy for other genetic alterations are necessary.

Comment 2. I suggest that you include some information in order to complete the manuscript. Below you can find some works that could give useful ideas in expanding this part:

1- Accuracy of CT-Guided Core-Needle Biopsy in Diagnosis of Thoracic Lesions Suspicious for Primitive Malignancy of the Lung: A Five-Year Retrospective Analysis. Tomography. 2022;8(6):2828-2838. doi: 10.3390/tomography8060236.

2- CT-guided percutaneous core needle biopsy for small (≤20 mm) pulmonary lesions. Clin Radiol. 2013 Jan;68(1):e43-8. doi: 10.1016/j.crad.2012.09.008.

3- CCN Guidelines® Insights: Non-Small Cell Lung Cancer, Version 2.2023. J Natl Compr Canc Netw. 2023 Apr;21(4):340-350. doi: 10.6004/jnccn.2023.0020.

Reply 2: We reviewed articles that you suggested. It was helpful to us to reinforce our rationale of this study. We modified introduction and discussion using the articles.

Changes in the text:

(Introduction, page 2, line 67-69) Molecular tests, epidermal growth factor receptor (EGFR) mutation, anaplastic lymphoma kinase (ALK) rearrangement, and programmed death-ligand 1 (PD-L1) expression, are also investigated in patients with early-stage NSCLC.[8]

(Introcudtion, page 3, line 101-103) Given these suboptimal results, platinum-doublet chemotherapy is still preferred as an adjuvant therapy in NSCLC.[8]  

(Discussion, page 12, line 376-378) The recently updated guideline recommend investigation of EGFR mutation status and ALK rearrangement for adjuvant therapy in resectable NSCLC.[8]

Comment 3. Please, improve the description of study aim.

Reply 3: We improved the description of study aim.

Changes in the text: (Introduction, page 3, line 106-112) Despite much progress during last decade, including targeted therapy and immunotherapy in advanced stage, the prognosis of NSCLC remains dismal. Therefore, more information is needed on patients with early-stage disease, especially in Asian populations, which tend to have a large number of non-smokers. In this study, we investigated the prevalence and importance of genetic alteration in patients with early-stage NSCLC who underwent definitive surgery in a single regional cancer center of Korea.

Comment 4. Please, underline the novelty of the study and ameliorate the description of possible clinical implications.

Reply 4. We modified the description of conclusion by referring to the comments of other reviewers as well.

Changes in the text: (Conclusion, page 13-14, line 411-423) In the present study, we showed that the prevalence of EGFR mutation, ALK rearrangement, and ROS1 fusion was 43.0%, 5.7%, and 1.6% in patients with NSCLC who underwent complete resection, respectively. EGFR mutation was an independent risk factor for recurrence along with stage II/III and solid pathologic subtype and was associated with a high rate of distant metastasis and a higher risk of CNS recurrence compared to wild-type EGFR. Among patients with recurrent malignancy, the rate of distant metastasis was higher in patients with EGFR mutations than in patients without EGFR mutations, and EGFR mu-tations were associated with CNS recurrence. Therefore, adjuvant or neoadjuvant targeted therapy could be considered more actively in patients with EGFR mutation who have undergone curative surgery. Further studies on perioperative therapy for other genetic alterations are also needed, and it is important to establish more sensitive methods, such as liquid biopsy, to predict recurrence and provide optimal therapy.

Reviewer 3 Report

Comments and Suggestions for Authors

The Authors present the results of a retrospective, monocentric study investigating the genetic alterations in a quite large coohrt of patients with early-stage NSCLC and the association between EGFR mutation and recurrence after complete resection.

The study is well conducted and methodologically sound, although with all the limitations of any retrospective study. However, I feel that the Authors should greatly clarify the novelty of the results, which seems overall confirmatory of existing knowledge.

Some other minor comments:

- the Simple Summary should be revised to make it... simpler

- line 46: maybe the word "oncological" is missing somewhere? I feel that the major cause of death worldwide remains CV disease.

- quality of figures should be improved

Comments on the Quality of English Language

Minor polishing required

Author Response

[Reviewer 3`s comments]
The Authors present the results of a retrospective, monocentric study investigating the genetic alterations in a quite large coohort of patients with early-stage NSCLC and the association between EGFR mutation and recurrence after complete resection.

Comment 1. The study is well conducted and methodologically sound, although with all the limitations of any retrospective study. However, I feel that the Authors should greatly clarify the novelty of the results, which seems overall confirmatory of existing knowledge.

Reply 1: Thank you for your valuable comments. Much progress has been achieved during last decade including immunotherapy and targeted therapies in advanced NSCLC. But the overall prognosis of NSCLC still dismal. So, we conducted this study to get more information of early-stage NSCLC especially in Korean with a large number of non-smoker who might be have driver mutations. This might be helpful to improve the prognosis of NSCLC who underwent definitive surgery.

Changes in the text: (Introduction, page 3, line 106-112) Despite much progress during last decade, including targeted therapy and immunotherapy in advanced stage, the prognosis of NSCLC remains dismal. Therefore, more information is needed on patients with early-stage disease, especially in Asian populations, which tend to have a large number of non-smokers. In this study, we investigated the prevalence and importance of genetic alteration in patients with early-stage NSCLC who underwent definitive surgery in a single regional cancer center of Korea.

Comment 2. Make Simple Summary simpler.

Reply 2: We summarized our results in the simple summary more simply.

Changes in the Simple Summary: Here, we investigated the prevalence of genetic alteration and its association with EGFR mutation and prognosis in early-stage non-small cell lung cancer (NSCLC) after curative resection. The results showed that the prevalence of EGFR mutation, ALK rearrangement, and ROS1 fusion was 43.0%, 5.7%, and 1.6%. Patients with EGFR mutant NSCLC had a higher risk of recurrence than those without EGFR mutation. Additionally, EGFR mutation was related to a high proportion of distant metastasis and a higher risk of central nervous system recurrence.

Comment 3. Line 46: maybe the word "oncological" is missing somewhere? I feel that the major cause of death worldwide remains CV disease.

Reply 3: Thank you for your detailed comment. We noticed that we missed the word. So, we add the word ‘cancer-related’

Changes in the text: (Introduction, page 2, line 50-51) In 2020, lung cancer was reported as the leading cause of cancer-related death worldwide and the second most commonly diagnosed cancer.[1]

Comment 4. quality of figures should be improved.

Reply 4: We received comments about the quality of figures from other reviewers. So, we checked the quality and alignment of figures.

Changes in the text: We submitted a revised figure 3 and figure 4 with colors.

Round 2

Reviewer 1 Report

Comments and Suggestions for Authors

No further comments.

Comments on the Quality of English Language

Minor editing required.

Reviewer 2 Report

Comments and Suggestions for Authors

The manuscript has been improved as requested, I have no further comments

Comments on the Quality of English Language

Minor editing of English language required

Reviewer 3 Report

Comments and Suggestions for Authors

thanks for having addressed my comments

Comments on the Quality of English Language

Minor editing required, but this can be done at the proofs stage.